# Optimization of the Heat Dissipation Performance of a Lithium-Ion Battery Thermal Management System with CPCM/Liquid Cooling

**DOI:** 10.3390/ma15113835

**Published:** 2022-05-27

**Authors:** Xiaoping Zeng, Zhengxing Men, Fang Deng, Cheng Chen

**Affiliations:** 1Engeneering Training Center, Chengdu Aeronautic Polytechnic, Chengdu 610021, China; zeng_8701@163.com; 2School of Aeronautical Manufacturing Industry, Chengdu Aeronautic Polytechnic, Chengdu 610021, China; amen1980@163.com; 3School of Mechatronics & Vehical Engineering, Chongqing Jiaotong University, Chongqing 400047, China; dengfang_cqjtu@163.com

**Keywords:** phase change material, liquid cooling, thermal management system, heat dissipation, lithium-ion battery

## Abstract

In view of the harsh conditions of rapid charging and discharging of electric vehicles, a hybrid lithium-ion battery thermal management system combining composite phase change material (PCM) with liquid cooling was proposed. Based on the numerical heat transfer model, a simulation experiment for the battery thermal management system was carried out. Taking the maximum temperature and temperature difference of the battery module as the objectives, the effects of PCM thickness, the liquid flow rate and the cross-sectional area of the liquid channel on the temperature of the battery module were analyzed using response surface methodology (RSM). The results show that 31 groups of candidate parameter combinations can be obtained through response surface analysis, and phase change material (PCM) thickness should be minimized in order to improve space utilization in the battery module. The optimal parameter combination is a flow rate of 0.4 m/s and a PCM thickness of 5.58 mm, with the cross-sectional area of the liquid channel as 3.35 mm^2^.

## 1. Introduction

The problems of global energy consumption and environmental pollution are becoming increasingly serious. Among the causal factors is the heavy use of fossil fuel vehicles. Electric vehicles (EVs), with advantages including cleanliness, zero pollution, and high energy conversion efficiency, have developed rapidly in recent years [1]. Lithium-ion batteries are the core components for energy storage in EVs, and their quality has a direct effect on the performance of EVs. Generally, the operating temperature of lithium-ion batteries ranges from 20 to 50 °C and the temperature difference must be controlled to within 5 °C [2,3]. Too high or low a temperature can significantly affect the performance and service life of batteries. To ensure the high efficiency and safe operation of EVs, a well-designed battery thermal management system (BTMS) is particularly important.

At present, heat dissipation methods for lithium-ion batteries in EVs mainly include air cooling, liquid cooling, heat pipe cooling and phase change cooling [4]. While air cooling has the advantage of simple structures and low cost, liquid cooling has higher thermal conductivity. However, with the increase in energy density of lithium-ion power batteries, conventional cooling methods such as air cooling and liquid cooling are no longer able to meet the heat dissipation requirements for battery packs in EVs under the stringent conditions of high-rate battery discharge and high ambient temperature. To further optimize the heat dissipation performance of the BTMS, researchers have presented different coupled heat dissipation systems, e.g., heat pipe/liquid cooling [5], heat pipe/air cooling [6], phase change material (PCM)/air cooling [7], and PCM/liquid cooling [8]. Compared with other coupled heat dissipation methods, the coupled heat dissipation method of composite phase change material (CPCM)/liquid cooling can combine advantages such as zero energy consumption of PCM and a high heat transfer coefficient to realize better uniformity of heat dissipation and faster thermal conductivity. Compared with the conventional water cooling heat dissipation method, the hybrid CPCM/liquid cooling BTMS can simplify the water cooling system, improve system reliability, and reduce the consumption of external energy, with a promising development prospect.

Bai et al. [9] put forward a coupled heat dissipation method of PCM/water-cooled plate water cooling for batteries and used the numerical simulation method for analysis, and the results showed that the PCM/water-cooled plate can effectively limit the maximum temperature of a battery after five consecutive charge–discharge cycles, prevent thermal runaway and effectively enhance the uniformity of batteries. Zhang et al. [10] proposed a new coupled BTMS that was based on PCM and liquid cooling. The performance of heat dissipation was improved through various parameters such as the thermal conductivity of optimized PCM and water flow velocity. The results indicated that this system incorporated the advantages of PCM and liquid BTMS and could fully meet the demand for heat dissipation under extreme operating conditions. Akbarzadeh et al. [11] designed the heat dissipation structure embedding PCM in the liquid cooled plate (LCP) for prismatic lithium-ion batteries and analyzed the performance of the heat dissipation of this structure. The results showed that the coupled thermal management system of PCM/LCP could not only reduce energy consumption but also improve the uniformity of battery temperature if the heat dissipation effect was the same. Cao et al. [12] put forward a delayed liquid cooling method combining PCM and liquid cooling for a module with 46 cylindrical batteries. The simulation results showed that when the battery discharged at 4C, the water flow was adjusted to control the battery temperature at 55 °C or below, and the delayed cooling control was conducted to improve the uniformity of battery temperature reduce the power consumption of the system. Song et al. [13] suggested a heat dissipation structure of coupled PCM/liquid cooling, replaced the cylindrical battery with a heating element, and researched the impact of geometrical parameters of the heat dissipation structure (e.g., thermal column size, thickness of heat dissipation plate, and battery spacing) on the heat dissipation effect. An et al. [14] proposed a coupled thermal management system integrating CPCM/liquid cooling and containing an aluminum frame for the 18650 cylindrical lithium-ion battery module and researched the effects of arrangement of the liquid channels, the liquid flow rate, the mass fraction of EG and discharge rate on the performance of the heat dissipation of a battery module. The results showed that the coupled thermal management system integrating CPCM/liquid cooling had better performance of heat dissipation at the discharge rate of 3C. Ling et al. [15] researched the performance of the heat dissipation of the coupled thermal management system integrating CPCM/liquid cooling for cylindrical lithium-ion batteries based on response surface methodology (RSM). This thermal management system could effectively reduce the PCM volume when the requirements for heat dissipation were met. Ping et al. [16] put forward a new BTMS of the coupled heat dissipation of PCM and liquid channels for prismatic lithium-ion power batteries. The simulation results showed that this system still had good cooling performance even if the ambient temperature was 45 °C, and the maximum temperature and temperature difference of a battery module are controlled at 47.6 and 4.5 °C, respectively; in the process of charge and discharge cycle, the liquid flow rate could be appropriately lowered, so that the power consumption of liquid cooling could be saved and the cooling capacity maintained.

It can be concluded from the above literature that although many contributions on the hybrid cooling method integrating CPCM/water cooling have been achieved, battery heat dissipation research was carried out at a discharge rate of 1C or 2C. On the other hand, some BTMSs proposed in the above literature with complex structure and low space utilization are not very suitable for practical application. Under harsh operation conditions such as acceleration, heavy load or climbing, the discharge rate of the battery will increase greatly compared to under normal driving conditions. At present, the research on the heat dissipation of lithium-ion batteries at a high discharge rate of 3C is still very insufficient for prismatic lithium-ion batteries due to lack of in-depth research on new heat dissipation structures and materials, coupled with the limitation of experimental conditions. Hereby, a serpentine micro-channel coupled BTMS combining CPCM/water cooling for prismatic lithium-ion batteries is proposed which is characterized by high space utilization and a simple structure, with research into the performance of heat dissipation at an ambient temperature of 40 °C and at the discharge rate of 3C.

## 2. Model and Methodology

### 2.1. Geometric Model

Figure 1 illustrates the mesh model of a battery module. Ten single prismatic lithium-ion batteries are arranged in parallel, the BTMS adopts the coupled heat dissipation method combining CPCM/liquid cooling, and the serpentine liquid flow channel is embedded in the 6 mm CPCM heat dissipation plate. The overall dimensions of a prismatic lithium-ion battery are 20 mm × 135 mm × 218 mm, with a capacity of 50 Ah and a nominal voltage of 3.6 V. The specific parameters are listed in Table 1 (supplied by Hengyu Technology Group, Shenzhen, China).

### 2.2. Numerical Model

The heat generation rate model proposed by Bernardi et al. [17] is used:(1)$$\;q=\frac{1}{V_{b}}\left[\left(E_{0}-U_{1}\right)-T\frac{dE_{0}}{dT}\right]$$
where *V_b_*, *E*_0_, and *U*_1_ are the volume of a single battery, the open-circuit voltage and the terminal voltage of a single battery, respectively; *I* is the current of battery charge and discharge; *T* is the thermodynamic temperature; $\frac{dE_{0}}{dT}$ is the coefficient of temperature effect. *E*_0_ − *U*_1_ can be replaced with the product of ohmic internal resistance (*R*_0_) and current intensity (*I*^2^) of a battery to obtain the heat generation rate of a single battery.
(2)$$\left(E_{0}-U_{1}\right)=I^{2}R_{0}$$
(3)$$q=\frac{1}{V_{b}}\left[I^{2}R_{0}-T\frac{dE_{0}}{dT}\right]$$

The CPCM is mixed by paraffin RT44HC and graphite, with a specific heat capacity of 1926 kJ·kg^−1^·K^−1^, a phase change latent heat of 258.5 kJ·kg^−1^, a thermal conductivity of 1.23 W·m^−1^·K^−1^ and a phase change point of 44 °C.

The enthalpy change is used to calculate the heat transfer of PCM, and the energy equation is as follows:(4)$$\rho_{\mathrm{PCM}}\frac{\partial H}{\partial t}=k_{PCM}\nabla^{2}T$$
(5)$$H=\beta L+\int_{T_{i}}^{T}cdT$$
(6)$$\beta =\left\{\begin{array}{c} 0\;\;\;\;\;\;\;\;\;\;T<T_{m}\\ \frac{T-T_{m}}{T_{m}-T_{1}}\;\;\;T_{m}<T<T_{1}\;\;\\ 1\;\;\;\;\;\;\;\;\;\;\;T>T_{1} \end{array}\right.$$
where $\rho_{\mathrm{PCM}}$ is the density of PCM; $k_{PCM}\;$ is the thermal conductivity of PCM; *β* is the liquid-phase volume fraction, *L* is the phase change latent heat, and *β*L is the enthalpy heat value of melted PCM; $\int_{T_{1}}^{T}cdT$ is the enthalpy value of sensible heat, *T_i_* is the initial temperature, and *c* is the specific heat capacity; *T_m_* is the temperature of melting point of PCM, and *T*_1_ is the temperature of PCM at the time of complete melting.

The following assumptions are made for the numerical simulation: (1) the impact of thermal radiation on the temperature field of the battery module is not considered; (2) the impact of supporting parts and other electronic components in the battery module is ignored in the simulation process; (3) the fluid is incompressible.

The initial temperature of the coolant and the ambient temperature are 40 °C, and the heat dissipation condition on the surface of the module is natural convection. The boundary conditions of the liquid inlet and outlet are set as the velocity inlet and the pressure outlet. The cooling medium is water. The above boundary conditions were set by commercial codes ANSYS/FLUENT.

### 2.3. Response Surface Model

The RSM is adopted to optimize and analyze the results of heat dissipation simulation analysis of a prismatic lithium-ion power battery module so as to obtain the change rule of maximum temperature and maximum temperature difference caused by multiple factors and obtain the optimal parameters. Three influencing factors are optimized in this study: (1) for the passive heat dissipation component, the optimal PCM thickness is selected; (2) for the cross-sectional area of the pipes, the optimal dimensions are selected; (3) for the flow rate, the optimal flow rate of heat dissipation is selected to reduce energy consumption. The BTMS shall be optimized and the performance of heat dissipation ensured; and for the purpose of optimization, the restrictive conditions are as follows: T_max_ < 50 °C, ΔT < 5 °C, and *β* ≤ 0.01.

The RSM adopts the Box-Behnken design method to build a mathematical model [18,19,20], and the parameters of coupled heat dissipation model integrating CPCM/water cooling are selected and optimized at three factors and three levels. Three parameters (i.e., the water flow rate A, the CPCM thickness B and the cross-sectional area of the liquid channel C) are selected and used as the three factors, and the maximum temperature, T_max_, and maximum temperature difference, ΔT, are used as the responses. The nonlinear fitting of the multivariate quadratic regression equation can be conducted, and the format of its fitting formula is as follows:(7)$$y=a_{0}+\sum_{i=1}^{3}a_{i}x_{i}+\sum_{i=1}^{3}a_{ii}x_{i}^{2}+\sum_{i,j=1}^{3}a_{ij}x_{i}x_{j}+\varepsilon \;\left(i\neq j\right)$$
where y is the system response value (T_max_ or ΔT), *x* is the variable (A, B or C), and *a* is the coefficient obtained through fitting.

## 3. Results and Discussion

### 3.1. RSM Analysis

The response surface experiment design is shown in Table 2 and the simulation experiment results are listed in this table.

The data obtained from the above-mentioned experiment are used for the quadratic response surface regression analysis to obtain the ternary quadratic response equation of T_max_ and ΔT of the battery module, as follows:T_max_ = 48.59 − 0.22A − 0.58B − 0.23C + 0.19AB + 0.072AC + 0.19BC + 0.013A^2^ + 0.41B^2^ + 0.077C^2^(8)
ΔT = 4.65 − 0.15A − 0.62B − 085C + 0.15AB + 0.055AC + 0.24BC − 0.028A^2^ + 0.39B^2^ − 0.02C^2^(9)

It can be seen from Equations (8) and (9) that the *p*-value of both of the two models is less than 0.05, indicating that the significance is significant.

### 3.2. Significance Analysis of Factors of T_max_

Based on the 17 groups of test results above, T_max_ of the battery module is analyzed to determine the impact of the liquid flow rate, CPCM thickness and the cross-sectional area of the liquid channel on T_max_ of the lithium-ion module at a discharge rate of 3C.

It can be seen from the analysis of variance in Table 3 that the *p*-value of the liquid flow rate, CPCM thickness and the cross-sectional area of the pipe is less than 0.0001, indicating that the impact on T_max_ of batteries is highly significant. The *p*-value of the model is also less than 0.0001, indicating that this model is extremely significant, so this model can be used to accurately predict the changes in T_max_ of the battery module.

It can be observed from Table 3 that the effect of the liquid flow rate, CPCM thickness and the cross-sectional area of the water channel on T_max_ of the battery module is significant, and the sequence of effects of these three factors is as follows: CPCM thickness > cross-sectional area of the liquid channel > the liquid flow rate. It can be found that CPCM can significantly reduce the maximum temperature of the battery because it can absorb a large amount of the heat generated by the battery during the phase transition in a timely manner.

It can be seen from the *p*-values of AB, AC and BC that there is some interaction among the three factors influencing T_max_ of batteries, and the specific interactive responses are shown in Figure 2, Figure 3 and Figure 4.

It can be seen from Figure 2 that T_max_ of the battery module decreases with an increase in the liquid flow rate and CPCM thickness. This is because as CPCM thickness increases, the CPCM can absorb more heat transferred from the battery during its transition from solid to liquid. With the increase in the liquid flow rate, more heat can be carried away from the CPCM and battery, and the maximum temperature of the battery module decreases.

It can be seen from Figure 3 and Figure 4 that T_max_ of the battery module decreases with an increase in the liquid flow rate, the cross-sectional area of the liquid channel and CPCM thickness. This is because with the increase in the cross-sectional area of the water channel and the flow rate, the volume of coolant through the CPCM per unit time increases, which can take away more heat stored in the CPCM. At the same time, with CPCM thickness, more heat is absorbed from the battery, thus reducing the maximum temperature of the battery module.

Through the optimization of response surface, the optimal combination of factors is as follows: the liquid flow rate is 0.341 m/s, the CPCM thickness is 5.7 mm, and the cross-sectional area of the pipe is 4.88 mm^2^; and in this case, T_max_ of the battery module is 48.3332 °C.

### 3.3. Significance Analysis of Factors of ΔT

Table 4 is the table of analysis of the variance of ΔT, in which the temperature difference for each of the 17 groups of tests ranges from 4 to 5 °C. The *p*-value of this model is far less than 0.01, indicating that this model is extremely significant. Based on the F-value, the degree of impact of various factors on the experiment results is compared, and the sequence of these factors on ΔT of batteries is as follows: CPCM thickness > the liquid flow rate > the cross-sectional area of the liquid channel.

Figure 5, Figure 6 and Figure 7 are the contours and response surface of the ΔT of batteries versus the three factors. In Figure 5 and Figure 7, the response surface is relatively steep, indicating that the interaction is significant; in Figure 6, the response surface slope is gentle, indicating that the interaction is not significant. It can be seen from Figure 5 that the ΔT of a battery module decreases with an increase in the liquid flow rate and CPCM thickness. This is due to the fact that as CPCM thickness increases, the CPCM can absorb more heat transferred from the battery during its transition from solid to liquid. At the same time, more battery heat can be carried away by the faster flowing water, so the battery temperature is more uniform.

It can be seen from Figure 6 that ΔT of a battery module decreases with an increase in the liquid flow rate and the cross-sectional area of the liquid channel. With the increase in the cross-sectional area of water channel and the flow rate, the volume of coolant through the CPCM per unit time increases, which can take away more heat stored in the CPCM, thus reducing the maximum temperature and temperature difference of the battery module. It can be seen from Figure 7 that the above-mentioned rule can be observed.

Through the optimization of response surface, the optimal combination of factors is as follows: the liquid flow rate is 0.336 m/s, the CPCM thickness is 5.9 mm, and the cross-sectional area of the pipe is 3 mm^2^; and in this case, ΔT of the battery module of this model is 4.224 °C.

### 3.4. Determination of the Optimal Combination Based on T_max_ and ΔT

The optimal combination of the maximum temperature of the battery module is as follows: the liquid flow rate is 0.341 m/s, the CPCM thickness is 5.7 mm, and the cross-sectional area of the liquid channel is 4.88 mm^2^; the optimal combination of the ΔT is as follows: the liquid flow rate is 0.336 m/s, the CPCM thickness is 5.9 mm, and the cross-sectional area of the liquid channel is 3 mm^2^. The values of the three factors at the optimal T_max_ and ΔT are not unified, the RSM is employed for the comprehensive optimization analysis of T_max_ and ΔT of the battery module, and the candidates of the battery module are shown in Table 5.

It can be seen from Table 5 that in all candidates of the battery module, the Candidate Combination 26 has the thinnest PCM, which can make the battery module more compact and lighter. Hereby, Candidate Combination 26 is selected as the optimal factor combination. When the cross-sectional area of the liquid channel is 3.35 mm^2^, the contours and response surface of two factors (the liquid flow rate and CPCM thickness) versus the desirability of heat dissipation performance of the battery module are shown in Figure 8.

It can be seen from Figure 8 that T_max_ and ΔT of the battery module decrease with an increase in the liquid flow rate and CPCM thickness. Through the optimization of response surface, the ΔT is predicted as 4.24353 °C and T_max_ as 48.3505 °C; and in this case, the liquid flow rate is 0.4 m/s, the CPCM thickness is 5.58 mm, and the cross-sectional area of the liquid channel is 3.35 mm^2^. To verify the predicted results, the numerical simulation was performed for the above-mentioned factor combinations, and the temperature distribution contours is shown in Figure 9; it can be seen that T_max_ of the battery module is 48.368 °C and ΔT is 4.299 °C, which meet the temperature requirement of the battery operation. Compared with the predicted values, the ΔT of the battery module is increased by 0.05547 °C and the maximum temperature is decreased by 0.0175 °C, and the deviation is very small, indicating that the prediction of the heat dissipation performance of the batteries based on this model is relatively accurate.

## 4. Conclusions

The hybrid thermal management system integrating CPCM/liquid cooling was put forward for prismatic lithium-ion batteries, and the simulation of the heat dissipation of a battery module was conducted to analyze the distribution of temperature fields of the battery module at an ambient temperature of 40 °C and a discharge rate of 3C. The impact of CPCM thickness, the liquid flow rate and the cross-sectional area of the liquid channel on the maximum temperature and maximum temperature difference of the battery module was determined, and the conclusions are as follows:

(1) When the liquid flow rate is 0.341 m/s, the CPCM thickness is 5.7 mm, and the cross-sectional area of the liquid channel is 4.88 mm^2^, the maximum temperature of the battery module is at the optimal value, i.e., 48.333 °C.

(2) When the liquid flow rate is 0.336 m/s, the CPCM thickness is 5.9 mm, and the cross-sectional area of the liquid channel is 3 mm^2^, the maximum temperature difference of the battery module is 4.224 °C.

(3) Given that PCM thickness is reduced to improve space utilization, the optimal parameter combination of the battery module is selected as follows: the liquid flow rate is 0.4 m/s, the CPCM thickness is 5.58 mm and the cross-sectional area of the liquid channel is 3.35 mm^2^.

This investigation mainly analyzed the heat dissipation performance of a CPCM/liquid-cooled composite battery thermal management system using a CFD simulation method. In further research, the effectiveness of this system can be verified through physical experiments.

## Figures and Tables

**Figure 1 materials-15-03835-f001:**
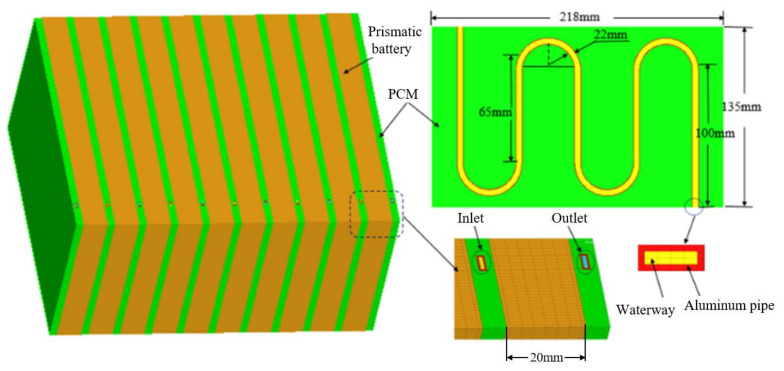
Mesh model of the battery module.

**Figure 2 materials-15-03835-f002:**
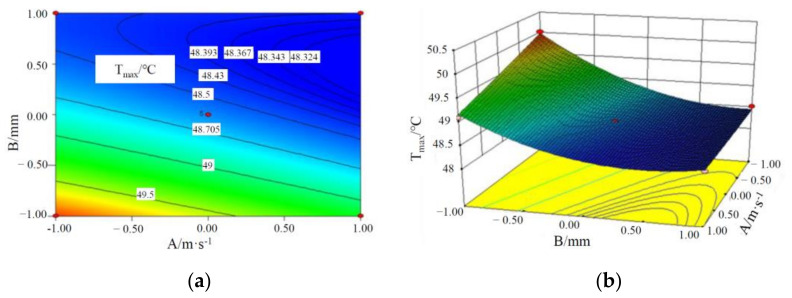
Contours and response surface of the impact of liquid flow rate and CPCM thickness on the maximum temperature. (**a**) Contours; (**b**) response surface.

**Figure 3 materials-15-03835-f003:**
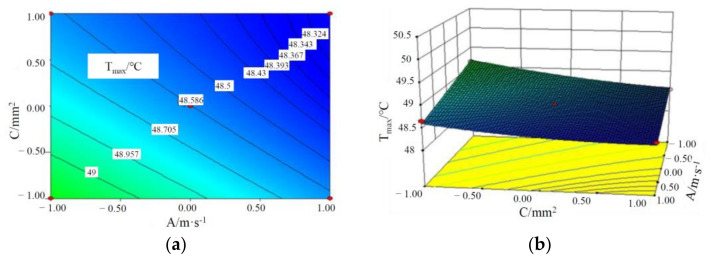
Contours and response surface of the impact of liquid flow rate and cross-sectional area of the pipe on the maximum temperature. (**a**) Contours; (**b**) response surface.

**Figure 4 materials-15-03835-f004:**
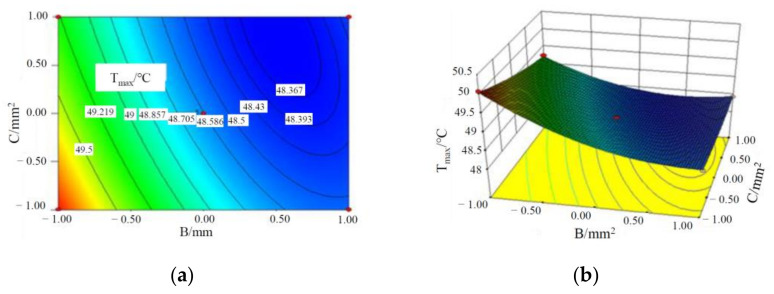
Contours and response surface of the impact of liquid flow rate and cross-sectional area of the pipe on the maximum temperature of a battery module. (**a**) Contours; (**b**) response surface.

**Figure 5 materials-15-03835-f005:**
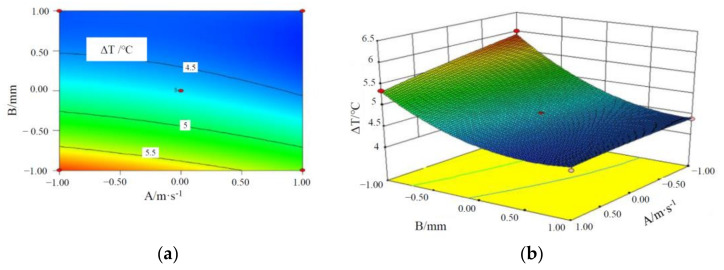
Contours and response surface of the impact of liquid flow rate and CPCM thickness on the maximum temperature difference. (**a**) Contours; (**b**) response surface.

**Figure 6 materials-15-03835-f006:**
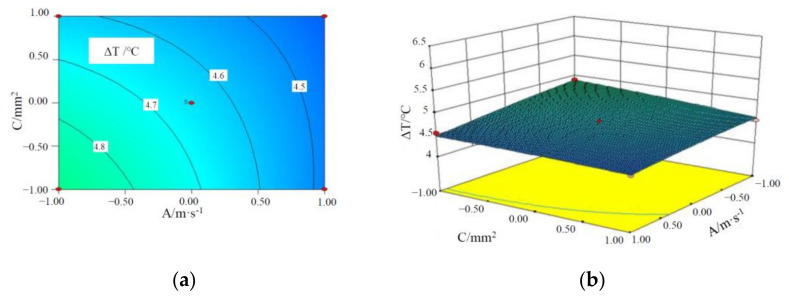
Contours and response surface of the impact of liquid flow rate and cross-sectional area of the pipe on the maximum temperature difference. (**a**) Contours; (**b**) response surface.

**Figure 7 materials-15-03835-f007:**
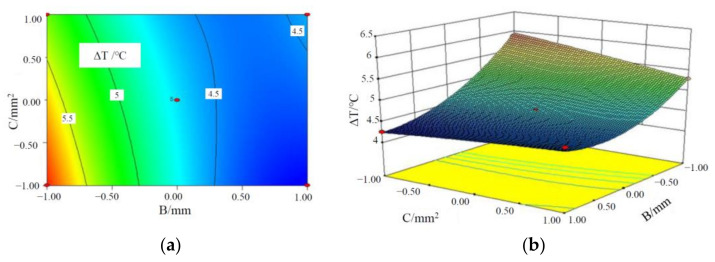
Contours and response surface of the impact of CPCM thickness and cross-sectional area of the pipe on the maximum temperature. (**a**) Contours; (**b**) response surface.

**Figure 8 materials-15-03835-f008:**
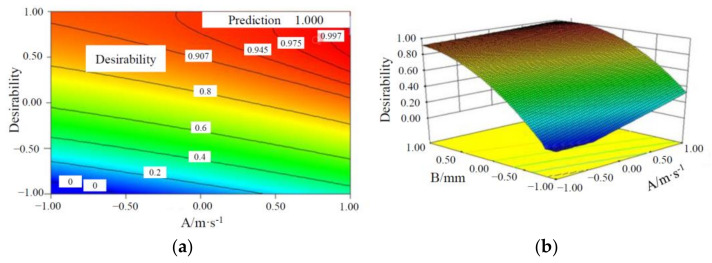
Contours and response surface of the optimal combination of factors versus the desirability of maximum temperature and maximum temperature difference. (**a**) Contours; (**b**) response surface.

**Figure 9 materials-15-03835-f009:**
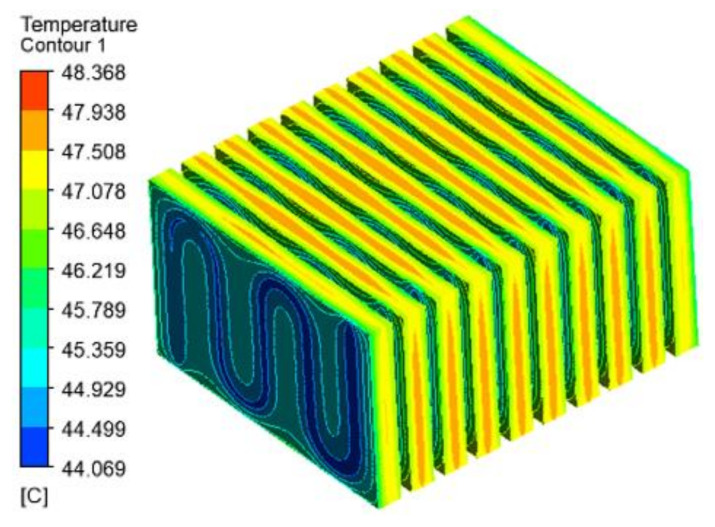
Temperature distribution contours of Candidate Combination 26.

**Table 1 materials-15-03835-t001:** Battery module parameters.

Parameter	Value	Unit
Density	2101	kg·m^−3^
Specific heat capacity	1014	kJ·kg^−1^·K^−1^
Radial thermal conductivity	0.813	W·m^−1^·K^−1^
Tangential and axial thermal conductivity	24.869	W·m^−1^·K^−1^

**Table 2 materials-15-03835-t002:** Experiment design and results.

Experiment No.	A /m·s^−1^	B /mm	C /mm^2^	T_max_ /°C	ΔT /°C
1	0.3	4	3	50.087	5.876
2	0.3	4	6	49.259	5.301
3	0.3	6	6	48.435	4.64
4	0.4	4	4	49.109	5.347
5	0.2	5	3	49.149	4.9
6	0.3	5	4	48.589	4.65
7	0.2	5	6	48.553	4.545
8	0.4	5	6	48.352	4.414
9	0.2	4	4	50.025	6.01
10	0.2	6	4	48.532	4.38
11	0.3	5	4	48.589	4.65
12	0.4	6	4	48.364	4.3
13	0.3	5	4	48.589	4.65
14	0.3	5	4	48.589	4.65
15	0.4	5	3	48.662	4.548
16	0.3	6	3	48.504	4.253
17	0.3	5	4	48.589	4.65

**Table 3 materials-15-03835-t003:** Analysis of the variance of T_max_.

Source	Quadratic Sum	Degrees of Freedom	Mean Square	F-Value	*p*-Value
Model	4.54	9	0.5	152.31	<0.0001
A	0.39	1	0.39	118.52	<0.0001
B	2.7	1	2.7	814.37	<0.0001
C	0.41	1	0.41	122.7	<0.0001
AB	0.14	1	0.14	42.24	0.0003
AC	0.02	1	0.02	6.17	0.0419
BC	0.14	1	0.14	43.49	0.0003
A^2^	0.00073	1	0.0007	0.22	0.654
B^2^	0.69	1	0.69	208.93	<0.0001
C^2^	0.025	1	0.025	7.51	0.0289
Residual error	0.023	7	0.0033		
Lack of fit	0.023	3	0.0077		
Pure error	0	4	0		
Total variation	4.56	16			

Notes: Extremely significant: *p* < 0.01; significant: *p* < 0.05; insignificant: *p* < 0.05.

**Table 4 materials-15-03835-t004:** Analysis of the variance of ΔT.

Source	Quadratic Sum	Degrees of Freedom	Mean Square	F-Value	*p*-Value
Model	4.54	9	0.48	85.23	<0.0001
A	0.9	1	0.19	33.65	0.0007
B	3.08	1	3.08	551.03	<0.0001
C	0.057	1	0.057	10.26	0.0150
AB	0.085	1	0.085	15.22	0.0059
AC	0.012	1	0.012	2.19	0.1827
BC	0.23	1	0.23	41.44	0.0004
A^2^	0.0034	1	0.0034	0.6	0.4633
B^2^	0.63	1	0.63	113.24	<0.0001
C^2^	0.0017	1	0.0017	0.3	0.5999
Residual error	0.039	7	0.0033		
Lack of fit	0.039	3	0.013		
Pure error	0	4	0		
Total variation	4.32	16			

Notes: Extremely significant: *p* < 0.01; significant: *p* < 0.05; insignificant: *p* > 0.05.

**Table 5 materials-15-03835-t005:** The candidates of the battery module.

Candidate Combination	A /m·s^−1^	B /mm	C /mm^2^	T_max_ /°C	ΔT /°C
1	1	0.75	−0.63	48.3516	4.22694
2	0.95	0.76	−0.57	48.3508	4.2444
3	0.93	0.67	−0.57	48.3475	4.25272
4	0.96	0.74	−0.55	48.3456	4.24805
5	1	0.69	−0.66	48.3507	4.22687
6	0.96	0.72	−0.60	48.3498	4.23904
7	0.95	0.72	−0.54	48.3436	4.25056
8	0.94	0.70	−0.57	48.3461	4.24965
9	0.98	0.67	−0.55	48.3389	4.25035
10	0.97	0.69	−0.59	48.3446	4.24248
11	0.93	0.69	−0.56	48.3472	4.25285
12	1	0.7	−0.59	48.3428	4.23756
13	1	0.81	−0.53	48.3498	4.24627
14	0.94	0.68	−0.58	48.3469	4.24821
15	0.94	0.77	−0.54	48.3498	4.25092
19	0.92	0.71	−0.58	48.3517	4.24886
17	1	0.77	−0.57	48.3472	4.23857
18	0.99	0.74	−0.57	48.3452	4.2403
19	0.97	0.74	−0.56	48.3454	4.24402
20	0.99	0.77	−0.52	48.3441	4.2487
21	1	0.81	−0.51	48.3478	4.2496
22	0.98	0.80	−0.54	48.3497	4.24487
23	0.96	0.66	−0.63	48.3499	4.24107
24	0.94	0.73	−0.55	48.3472	4.25105
25	0.96	0.62	−0.58	48.3435	4.25264
26	1	0.58	−0.65	48.3505	4.24353
27	0.93	0.68	−0.58	48.3479	4.24977
28	0.95	0.6	−0.62	48.3512	4.25198
29	0.97	0.74	−0.60	48.3504	4.23793
30	1	0.75	−0.52	48.3412	4.24714
31	0.99	0.68	−062	48.3467	4.23574

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
