# Peer review of "Optimization of the Heat Dissipation Performance of a Lithium-Ion Battery Thermal Management System with CPCM/Liquid Cooling"

_materials, 2022, doi:10.3390/ma15113835_

Round 1

Reviewer 1 Report

This paper focuses on a hybrid lithium-ion battery thermal management system combining composite phase change material with liquid cooling to improve the space utilization of the battery module. Also, an optimal parameter combination is formulated and verified. The concept is interesting, the methodology is well presented, but the paper needs some major revisions.

Author Response

Thank you for your comments, which are highly insightful and enabled us to greatly improve the quality of the manuscript.  we extensively revised the manuscript and point-by-point responsed to each of the comments according to the comments. 

Reviewer 2 Report

Title: Optimization on Heat Dissipation Performance of Lithium-ion Battery…

Manuscript ID: materials-1700367

Authors: Xiao-ping et al.

Dear Authors,

Thank you for the opportunity to read your article. I found the topic is interesting and fundamental. Generally speaking, there are some results presented in order to capture some trends, but the introduction and methods need more clear explanation while the results need reorganization and discussion with fair point of view. I suggest that this article will be revised extensively before its re-submission for another review process if applicable. As a conclusion, I recommend its major revision at this state.

I hope my comments are helpful.

Good luck,

A reviewer

Major concerns:

“Keywords”

-Please consider listing keywords that are not used in the article title.

“1. Introduction”

-In Introduction, please consider clearly mentioning the research gap(s) you tried to address in this work. In other words, how and why “the research of the hybrid cooling method…is still very insufficient…”

-Lines 29-31: Please consider citing your reference(s) of those values (i.e. 15 to 50 ºC, 5 ºC temperature difference).

“2. Model and Methodology”

-Table 1: Please consider providing reference(s) for those parameters.

-Lines 119-121: Please consider combining those information with the ones on lines 114-117 if appropriate.

-Equation (7): Please consider revising it with A, B, C for a reader to understand your equation better and its correlation with the experimental variables shown in Table 2.

“3. Results and Discussion”

-Lines 145: “…the Box-Behnken design method…”->Please consider citing a reference.

-Table 2: Please consider reorganizing the way you list “Experiment No.”, for example from 1 to 17 as the current list does not have any clear rule of the organization.

-Lines 156-160: Please see my comment on Equation (7).

-Lines 169-173: Please consider splitting this long sentence into two for the clarification of your message. In addition, I am not sure why you mentioned “…R2 is 0.9949… R2adj is 0.9187…is better…”

-Figures 5-7: Please consider agglomerating them to formulate one figure for a reader to compare the results easily. In fact, on lines 225-232 you compare them.

-Table 5: “…the optimal heat dissipation design…”->Please consider revising the table title since it is strange to have 31 optimal design.

-Lines 245: Please consider mentioning the selection criteria of the best combination. In other words, please mention why you selected combination 26 the best.

-Line 260: “…Tmax…48.368 ºC and deltaT is 4.299 ºC.”->Please consider mentioning whether your results addressed the criteria mentioned on lines 29-31.

“4. Conclusion”-> 4. Conclusions

-You may state future perspectives in Conclusions.

Minor concerns:

-Line 245: “…Table 7…”->Table 5?

Author Response

(The authors gave the same response as above.)

Round 2

Reviewer 2 Report

Dear Authors,

As all the comments were addressed, I would suggest the journal accept this article for its publication.

Best regards,
A reviewer